# Oscillators that sync and swarm

Kevin P. O'Keeffe[1], Hyunsuk Hong [2] & Steven H. Strogatz [1]

Synchronization occurs in many natural and technological systems, from cardiac pacemaker cells to coupled lasers. In the synchronized state, the individual cells or lasers coordinate the timing of their oscillations, but they do not move through space. A complementary form of self-organization occurs among swarming insects, flocking birds, or schooling fish; now the individuals move through space, but without conspicuously altering their internal states. Here we explore systems in which both synchronization and swarming occur together. Specifically, we consider oscillators whose phase dynamics and spatial dynamics are coupled. We call them swarmalators, to highlight their dual character. A case study of a generalized Kuramoto model predicts five collective states as possible long-term modes of organization. These states may be observable in groups of sperm, Japanese tree frogs, colloidal suspensions of magnetic particles, and other biological and physical systems in which self-assembly and synchronization interact.

[1] Center for Applied Mathematics, Cornell University, Ithaca, NY 14853, USA. [2] Department of Physics and Research Institute of Physics and Chemistry, Chonbuk National University, Jeonju 561-756, Korea. Correspondence and requests for materials should be addressed to S.H.S. (email: strogatz@cornell.edu)

This year marks the fiftieth anniversary of a breakthrough in the study of synchronization. In 1967, Winfree proposed a coupled oscillator model for the circadian rhythms that underlie daily cycles of activity in virtually all plants and animals[1]. He discovered that above a critical coupling strength, synchronization breaks out spontaneously, in a manner reminiscent of a phase transition. Then Kuramoto simplified Winfree's model and solved it exactly[2], leading to an explosion of interest in the dynamics of coupled oscillators[3–5]. Kuramoto's model in turn has been generalized to other large systems of biological oscillators, such as chorusing frogs[6], firing neurons[7–11], and even human concert audiences clapping in unison[12]. The analyses often borrow techniques from statistical physics, such as mean-field approximations, renormalization group analyses[13,14], and finite-size scaling[15,16]. There has also been traffic in the other direction, from biology back to physics. For example, insights from biological synchronization have shed light on neutrino oscillations[17], phase locking in Josephson junction arrays[18], the dynamics of power grids[19,20], and the unexpected wobbling of London's Millennium Bridge on opening day[21].

A similarly fruitful interplay between physics and biology has occurred in the study of the coordinated movement of groups of animals. Fish schools, bird flocks, and insect swarms[22–26] have been illuminated by maximum entropy methods[27], agent-based simulations[28], and analytically tractable models based on self-propelled particles[29], and continuum limits[30–33].

Studies of swarming and synchronization have much in common. Both involve large, self-organizing groups of individuals interacting according to simple rules. Both lie at the intersection of nonlinear dynamics and statistical physics. Nevertheless the two fields have, by and large, remained disconnected. Studies of swarms focus on how animals move, while neglecting the dynamics of their internal states. Studies of synchronization do the opposite: they focus on oscillators' internal dynamics, not on their motion. In the past decade, however, a few studies of "mobile oscillators," motivated by applications in robotics and developmental biology, have brought the two fields into contact[34–38]. Even so, the assumption has been that the oscillators' locations affect their phase dynamics, but not conversely. Their motion has been modeled as a random walk or as externally determined, without feedback from the oscillators' phases.

We suspect that somewhere in nature and technology there must be mobile oscillators whose phases affect how they move. For instance, many species of frogs, crickets, and katydids call periodically, and synchronize in vast choruses[6,39–41]. The natural question is whether they tend to hop toward or away from others depending on the relative phases of their calling rhythms, and if so, what spatiotemporal patterns are produced.

A clue comes from the physics of magnetic colloids[42–44] and microfluidic mixtures of active spinners[45,46], both of which show rich collective behavior. In these systems, the particles or spinners attract or repel one another, depending on their orientations. Given that orientation is formally analogous to the phase of an oscillation (both being circular variables), a similarly rich phenomenology is expected for mobile oscillators whose phases affect their motion. We call these hypothetical systems 'swarmalators' because they generalize swarms and oscillators.

One possible instance of a swarmalator system is a population of myxobacteria, modeled in 2001 by Igoshin and colleagues[47]. The movements of these bacteria in space are thought to be influenced by an internal, biochemical degree of freedom, which appears to vary cyclically. Igoshin et al.[47] modeled it as a phase oscillator. Experimental evidence suggests that the evolution of this phase is influenced by the spatial density of neighboring cells; thus there appears to be a bidirectional coupling between spatial and phase dynamics, as required of swarmalators.

Tanaka and colleagues also made an early contribution to the modeling of swarmalators[48,49]. They analyzed a broad class of models in the hope of finding phenomena which were not system-specific. They considered chemotactic oscillators, whose movements in space are mediated by the diffusion of a background chemical. The oscillators' consumption of this chemical depends on their internal states, thereby completing the bidirectional space-phase coupling. Tanaka et al.[48,49] began with a general model with these ingredients, from which they derived a simpler model by means of center manifold and phase-reduction methods.

Here we take a bottom-up approach. We propose a simple model of a swarmalator system which lets us study some of its collective states analytically. We hope our work will draw attention to this class of problems, and stimulate the discovery and characterization of natural and technological systems of swarmalators.

## Results

**The model.** We consider swarmalators free to move in the plane. The governing equations are

$$\dot{\mathbf{x}}_i = \mathbf{v}_i + \frac{1}{N} \sum_{j=1}^{N} \left[ \mathbf{I}_{\text{att}}(\mathbf{x}_j - \mathbf{x}_i) F(\theta_j - \theta_i) - \mathbf{I}_{\text{rep}}(\mathbf{x}_j - \mathbf{x}_i) \right], \quad (1)$$

$$\dot{\theta}_i = \omega_i + \frac{K}{N} \sum_{j=1}^{N} H_{\text{att}}(\theta_j - \theta_i) G(\mathbf{x}_j - \mathbf{x}_i) \quad (2)$$

for $i = 1, \dots, N$, where $N$ is the population size, $\mathbf{x}_i = (x_i, y_i)$ is the position of the $i$-th swarmalator, and $\theta_i$, $\omega_I$, and $\mathbf{v}_i$ are its phase, natural frequency, and self-propulsion velocity. The functions $\mathbf{I}_{\text{att}}$ and $\mathbf{I}_{\text{rep}}$ represent the spatial attraction and repulsion between swarmalators, while the phase interaction is captured by $H_{\text{att}}$. The function $F$ in Eq. (1) measures the influence of phase similarity on spatial attraction, while $G$ in Eq. (2) measures the influence of spatial proximity on the phase attraction.

Consider the following instance of this model:

$$\dot{\mathbf{x}}_i = \mathbf{v}_i + \frac{1}{N} \left[ \sum_{j \neq i}^{N} \frac{\mathbf{x}_j - \mathbf{x}_i}{|\mathbf{x}_j - \mathbf{x}_i|} \left( A + J \cos(\theta_j - \theta_i) \right) - B \frac{\mathbf{x}_j - \mathbf{x}_i}{|\mathbf{x}_j - \mathbf{x}_i|^2} \right]$$
$$(3)$$

$$\dot{\theta}_i = \omega_i + \frac{K}{N} \sum_{j \neq i}^{N} \frac{\sin(\theta_j - \theta_i)}{|\mathbf{x}_j - \mathbf{x}_i|}. \quad (4)$$

For simplicity, we chose power laws for $\mathbf{I}_{\text{att}}$, $\mathbf{I}_{\text{rep}}$, and $G$ along with analytically convenient exponents. The sine function in $H_{\text{att}}$ was similarly motivated, in the spirit of the Kuramoto model[2]. We first consider identical swarmalators so that $\omega_i = \omega$ and $\mathbf{v}_i = \mathbf{v}$. Further, we assume propulsion with constant magnitude and direction $\mathbf{v} = v_0 \hat{n}$ where $\hat{n}$ is a constant vector (we relax these simplifications later). Then by a choice of reference frame we can set $\omega = v_0 = 0$ without loss of generality. Finally, by rescaling time and space we set $A = B = 1$. This leaves us with a system with two parameters $(J, K)$.

The parameter $K$ is the phase coupling strength. For $K > 0$, the phase coupling between swarmalators tends to minimize their phase difference, while for $K < 0$, this phase difference is maximized. The parameter $J$ measures the extent to which phase similarity enhances spatial attraction. For $J > 0$, "like attracts like": swarmalators prefer to be near other swarmalators with the same phase. When $J < 0$, we have the opposite scenario: swarmalators

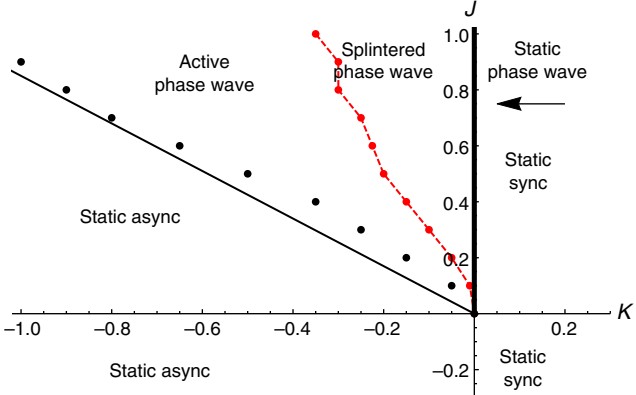

**Fig. 1** Phase diagram. Locations of states of the model defined by Eqs. (3) and (4) with $A = B = 1$ and $\mathbf{v}_i = \omega_i = 0$ in the $(J, K)$ plane. The straight line separating the static async and active phase wave states is a semi-analytic approximation given by (18). Black dots show simulation data. These were calculated by finding where the order parameter $S$ bifurcates from zero, defined by where its second derivative with respect to $K$ is largest. Similarly, the red dots separating the active phase wave and splintered phase wave states were found by finding where the order parameter $\gamma$ bifurcates from 0. The red dashed line simply connects these points and was included to make the boundary clearer

are preferentially attracted in space to those with opposite phase. And when $J = 0$, swaramalators are phase-agnostic, their spatial attraction being independent of their phase. To keep $\mathbf{I}_{att} > 0$, we constrain $J$ to satisfy $-1 \leq J \leq 1$.

Before stating our results, we pause to discuss our model's features. As mentioned above, the model's purpose is to study the interplay between synchronization and swarming. Let us clarify what we mean by swarming. While, to our knowledge, there is no unanimous classification, elements of a swarming system typically attract and repel each other, leading to aggregation, and align their orientations so as to move in the same direction. Succinctly then, a swarming system models aggregation and/or alignment.

Our model accounts for aggregation, but not for alignment: the spatial dynamics (1) model phase-dependent aggregation, while the phase dynamics (2) model position-dependent synchronization. There are no alignment terms. Indeed, the particles of our system do not have an orientation so there is nothing to align! We chose to neglect an orientation state variable, and thus alignment, for two reasons. The first was simply because we believe there are swarmalator systems in which orientation does not play a role, such as the Japanese tree frogs[6,41] or chemotactic oscillators[48,49]. The second was that modeling orientable swarmalators adds an additional layer of complexity; it gives each swarmalator an orientation $\beta$, increasing the number of state variables per swarmalator from three (a two-dimensional position $(x, y)$ and an internal phase $\theta$) to four.

In the interest of minimalism we wished to avoid this complication for now. Hence as it stands our model applies only to swarmalators without an orientation. However we later show that our results are robust to the inclusion of simple alignment dynamics, indicating their potential to hold for systems of orientable swarmalators as well.

**Numerics.** We performed numerical experiments to probe the behavior of our system. Unless otherwise stated, the simulations were run using python's ODE solver 'odeint'. We initially positioned the swarmalators in a box of length 2 and drew their phases from $[-\pi, \pi]$, both uniformly at random. We found the system settles into five states (Supplementary Movies 1–5). In

three of these states, the swarmalators are ultimately static in space and phase. In the remaining two, the swarmalators move. However in all states, the density of swarmalators $\rho(\mathbf{x}, \theta, t)$ is time-independent, where $\rho(\mathbf{x}, \theta, t) d\mathbf{x} \, d\theta$ gives the fraction of swarmalators with positions between $\mathbf{x}$ and $\mathbf{x} + d\mathbf{x}$, and phases between $\theta$ and $\theta + d\theta$ at time $t$. In Fig. 1 we show where these states occur in the $(J, K)$ parameter plane. In Figs. 2–6 we show their key properties. We next discuss these five states.

Static synchrony: The first state is shown in Fig. 2a. The swarmalators form a circularly symmetric, crystal-like distribution in space, and are fully synchronized in phase, as indicated by all of them having the same color in Fig. 2a. Since the swarmalators are ultimately stationary in $\mathbf{x}$, and they all end up at the same phase $\theta$, we call this the 'static sync' state. It occurs for $K > 0$ and for all $J$, as seen in Fig. 1.

In the continuum limit, this state is described by $\rho(r, \phi, \theta, t) = \frac{1}{2\pi} g_1(r)\delta(\theta - \theta_0)$, where $\phi$ is the spatial angle $\phi = \tan^{-1}(y/x)$, and the final phase $\theta_0$ is determined from the initial conditions. In Supplementary Note 1 we used a technique by Kololnikov et al.[50] when studying swarms to derive the following pair of integral equations for $g_1$:

$$\int_0^R \left[ (s-r)\mathcal{K}\left(\frac{4rs}{(r+s)^2}\right) + (r+s)\mathcal{E}\left(\frac{4rs}{(r+s)^2}\right) + \frac{\pi^2}{2J}(r-s) \right] \frac{2Js}{r} g_1(s) \, ds = 0 \tag{5}$$

$$g_1(r) = \frac{2(1+J)}{\pi} \int_0^R \mathcal{K}\left(\frac{4sr}{(r+s)^2}\right) \frac{g_1(s)}{s+r} s \, ds, \tag{6}$$

where $\mathcal{K}, \mathcal{E}$ are the complete elliptic integral of the first and second kinds, and $R$ is the radius of the disk in the $(x, y)$ plane which must be determined. We were unable to solve these equations for $g_1(r)$ and $R$, so instead solve them numerically, and show the results in Supplementary Note 1. Analytic progress can however be made if a linear attraction kernel $\mathbf{I}_{att}(\mathbf{x}) = \mathbf{x}$, is used instead of the unit vector kernel we are currently considering. Then, as shown in Kolokolnikov et al.[50], the radial density becomes $g_1(r) = 1$, i.e., swarmalators are uniformly distributed. In this special case we can also calculate $R$ analytically,

$$R_{sync} = (1+J)^{-1/2}. \tag{7}$$

We show a full derivation in the Methods section. In dimensionful units, this reads $R = \sqrt{B/(A+J)}$. Thus the radius is determined by the ratio of the strengths of the attractive to the repulsive forces $\mathbf{I}_{att}$, $\mathbf{I}_{rep}$ (in the static sync state, the effective attraction force is $A + J\cos(\theta_j - \theta_i) = A + J$, since all swarmalators have the same phase). Figure 4a shows the prediction (7) agrees with simulation results.

Static asynchrony: Swarmalators can also form a static async state, illustrated in Fig. 2b. At any given location $\mathbf{x}$, all phases $\theta$ can occur, and hence all colors are present everywhere in Fig. 2b. This is seen more clearly in a scatter plot of the swarmalators in the $(\phi, \theta)$ plane, depicted in Fig. 3a. Notice that the swarmalators are distributed uniformly, meaning that every phase occurs everywhere. This completely asynchronous state occurs in the quadrant $J < 0$, $K < 0$, and also for $J > 0$ as long as $J$ lies in the wedge $J < |K_c|$ shown in the phase diagram in Fig. 1. As for the static sync state, we were able to calculate the radius of the circular distribution when a linear attraction kernel $\mathbf{I}_{att}(\mathbf{x})$ was used. In the Methods section we show this radius is given by

$$R_{async} = 1 \tag{8}$$

which agrees with simulation as shown in Fig. 4a.

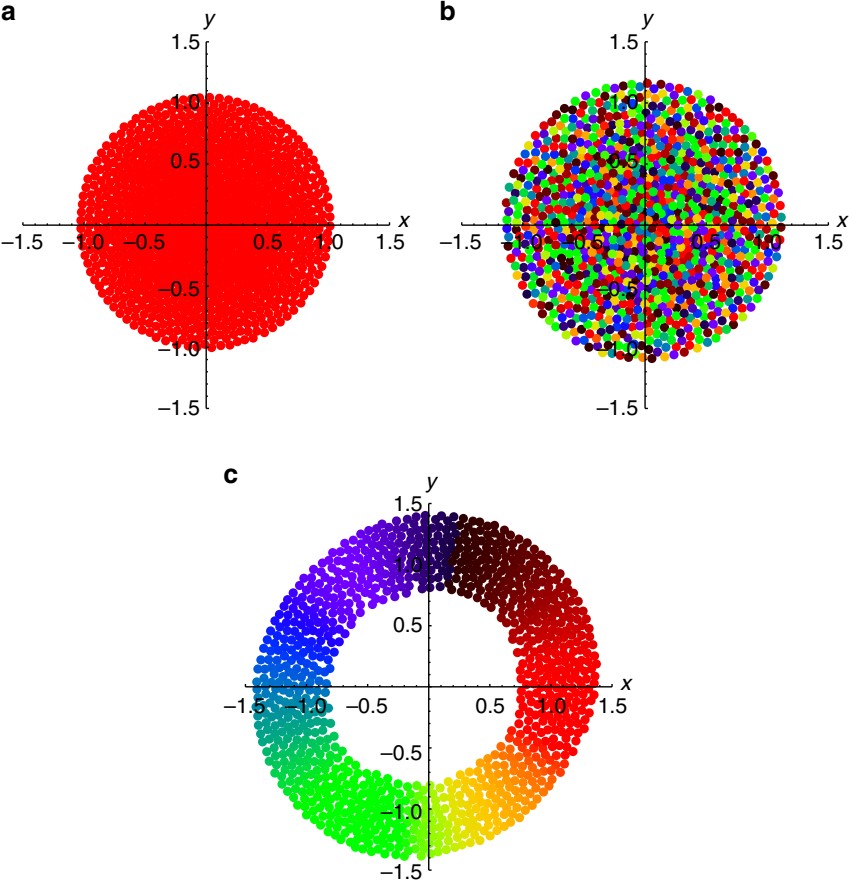

**Fig. 2** Stationary states. Scatter plots of three states in the (x, y) plane, where the swarmalators are colored according to their phase. Simulations were for N = 1000 swarmalators for T = 100 time units and stepsize dt = 0.1. Supplementary Movies 1–3 correspond to panels **a**–**c**. **a** Static sync state for (J, K) = (0.1, 1). **b** Static async state (J, K) = (0.1, −1). **c** Static phase wave state (J, K) = (1, 0)

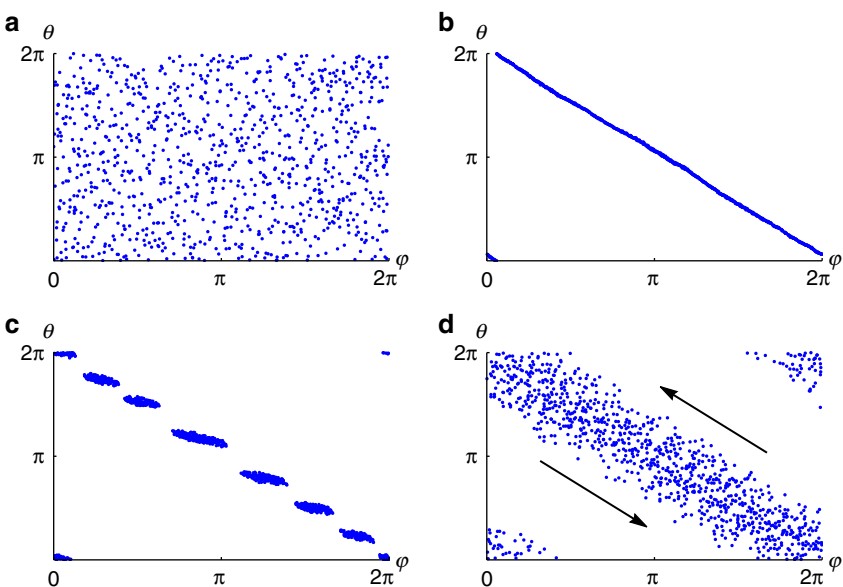

**Fig. 3** Scatter plots in ($\phi$, $\theta$) space. Distributions in ($\phi$, $\theta$) space corresponding to different states, where $\phi = \tan^{-1}(y/x)$. Simulations were run with N = 1000 swarmalators for variable numbers of time units T and stepsize dt = 0.1. **a** Static async state for (J, K) = (0.1, −1) and T = 100. **b** Static phase wave state (J, K) = (1, 0) and T = 100. **c** Splintered phase wave state (J, K) = (1, −0.1) and T = 1000. **d** Active phase wave state (J, K) = (1.0, −0.75) and T = 1000. Black arrows indicate the shear flow motion of swarmalators. Supplementary Movies 6 and 7 correspond to panels **c**, **d**

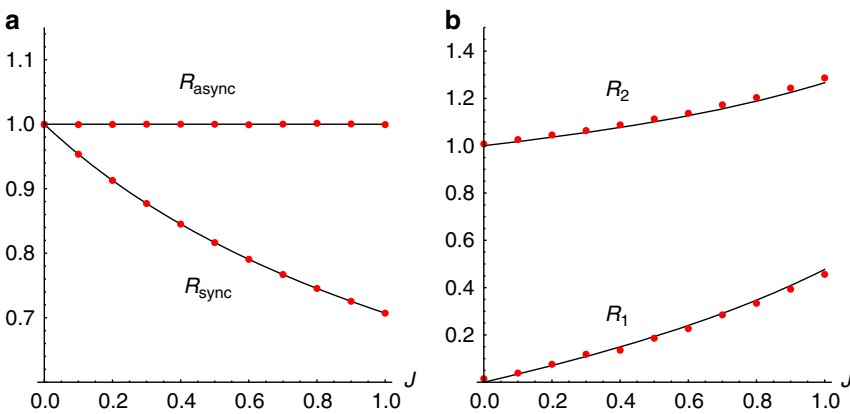

**Fig. 4** Radii of stationary states. Simulation were for 800 swarmalators with a linear attraction kernel $\mathbf{I}_{att}(\mathbf{x}) = \mathbf{x}$. Red dots show simulation data, while black curves show theoretical predictions. **a** Radius of crystal formed in static sync state (for $K = 1$) and static async state (for $K = -2$) vs. $J$. **b** Inner and outer radii of annulus in static phase wave state vs. $J$

Static phase wave: The final stationary state occurs for the special case $K = 0$ and $J > 0$. This means the swarmalators' phases are frozen at their initial values. Still, the population evolves by rearranging itself in space. Since $J > 0$, 'like attracts like': swarmalators want to settle near others with similar phase. The result is an annular structure where the spatial angle $\phi$ of each swarmalator is perfectly correlated with its phase $\theta$, as seen in Figs. 2c and 3b. Since the phases run through a full cycle as the swarmalators arrange themselves around the ring, we call this state the 'static phase wave'.

In density space, this static phase wave state is described by $\rho(r, \phi, \theta) = g_2(r)\delta(\phi \pm \theta + C_1)$ where the $\pm$ and the constant $C_1$, are determined by the initial conditions. In the Methods section we again consider the linear attraction kernel, and find that $g_2(r)$ can be obtained analytically,

$$g_2(r) = 1 - \frac{\Gamma_J}{r}, \quad R_1 \leq r \leq R_2 \quad (9)$$

with $\Gamma_J = 2J\left(R_2^3 - R_1^3\right)\left(3J\left(R_2^2 - R_1^2\right) + 12\right)^{-1}$. This in turn lets us calculate the inner and outer radii $R_1$, $R_2$ of the annulus:

$$R_1 = \Delta_J \frac{-\sqrt{3}J - 3\sqrt{12 - 5J}\sqrt{J + 4} + 12\sqrt{3}}{12J}, \quad (10)$$

$$R_2 = \frac{\Delta_J}{2\sqrt{3}} \quad (11)$$

with $\Delta_J = \sqrt{\frac{3J - \sqrt{36 - 15J}\sqrt{J+4} - 12}{J - 2}}$. Figure 3b shows agreement between these predictions and simulation.

Splintered phase wave: Moving from $K = 0$ into the $K < 0$ half-plane, we encounter the first non-stationary state, shown in Figs. 3c and 5a. As can be seen, the static phase wave splinters into disconnected clusters of distinct phases. Accordingly we call this state the 'splintered phase wave'. It is unclear what determines the number of clusters. Fewer are found when smaller length scales for the interaction functions $\mathbf{I}_{att}$, $\mathbf{I}_{rep}$, $G$ are used. However the parameters $J$, $K$ also play a role, although how precisely has not yet been determined. Within each cluster, the swarmalators "quiver," executing small amplitude oscillations in both position and phase about their mean values.

Active phase wave: As $K$ is further decreased, these oscillations increase in amplitude until the swarmalators start to execute regular cycles in both spatial angle and phase. This motion is best illustrated in Fig. 3d, in which shear flow about the $\phi_i = \theta_i \pm C$ axis is evident. This type of flow follows from a conserved

quantity in the model: $\langle \dot{\phi} \rangle = \langle \dot{\theta} \rangle = 0$, which can be seen by averaging Eqs. (3) and (4) over the population. There are also oscillations in the radial position, where each swarmalator travels from the inner rim to the outer rim and back, in one orbit around the annulus.

This new, and final, state is similar to the double milling states found in biological swarms[51], where populations split into counter-rotating subgroups. It is also similar to the vortex arrays formed by groups of sperm[52], where the angular position $\phi$ of each sperm is correlated with the phase $\theta$ associated with the rhythmic beating of its tail.

At the density level, the state is like a blurred version of the static phase wave, insofar as the spatial angle and phase of a given swarmalator are roughly correlated, as evident in Fig. 3d. However unlike the static phase wave, the swarmalators are non-stationary. To highlight this difference, we name this state the 'active phase wave'.

**Order parameters**. Having described the five states of our system, we next discuss how to distinguish them. We define the following order parameter,

$$W_\pm = S_\pm e^{i\Psi_\pm} = \frac{1}{N}\sum_{j=1}^N e^{i(\phi_j \pm \theta_j)}, \quad (12)$$

where $\phi_i := \tan^{-1}(y_i/x_i)$. As shown in Fig. 6, the magnitude $S_\pm$ varies from 1 to 0 as we decrease $K$ from 0, passing through all the states in the upper left quadrant of the $(J,K)$ plane. (Note that all states except for static sync occur in this part of parameter space, so we hereafter confine our attention to just this region.)

To see why $S_\pm$ varies in this manner, recall that in the static phase wave, the spatial angle and phase of each swarmalator are perfectly correlated, $\phi_i = \pm\theta_i + C_1$ (recall that the $\pm$ and $C_1$ are determined by the initial conditions. This means either $S_+$ or $S_-$ is non-zero). Therefore $S_\pm = 1$ at $K = 0$, where the static phase wave state is realized. Moving into the $K < 0$ plane we encounter the splintered phase wave. Here the correlation between $\phi_i$ and $\theta_i$ is not perfect, and so $S_\pm < 1$. As $K$ is decreased the decay of this correlation is non-monotonic, which induces a dip in $S_\pm$ as seen in Fig. 6. Once the active phase wave is reached however this non-monotonicity disappears. As a result $S_\pm$ declines uniformly until it finally drops to zero when the static async state is reached, in which $\phi_i$ and $\theta_i$ are fully uncorrelated.

To sum up, $S_\pm$ is zero in the static async state, bifurcates from zero at a critical coupling strength $K_c$, is non-zero in the non-

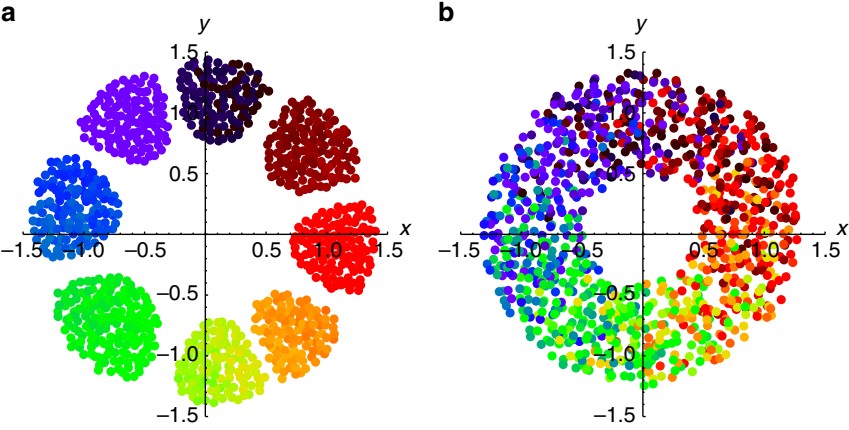

**Fig. 5** Non-stationary states. Simulation were run for $N = 1000$ swarmalators for $T = 1000$ time units and stepsize $dt = 0.1$. In all cases, swarmalators were initially placed in a box of length 2 uniformly at random, while their phases were drawn from $[-\pi, \pi]$. **a** Splintered phase wave $(J, K) = (1, -0.1)$. Note, there is a long transient until this state is achieved. See Supplementary Movie 4. **b** Active phase wave $(J, K) = (1, -0.75)$. See Supplementary Movie 5

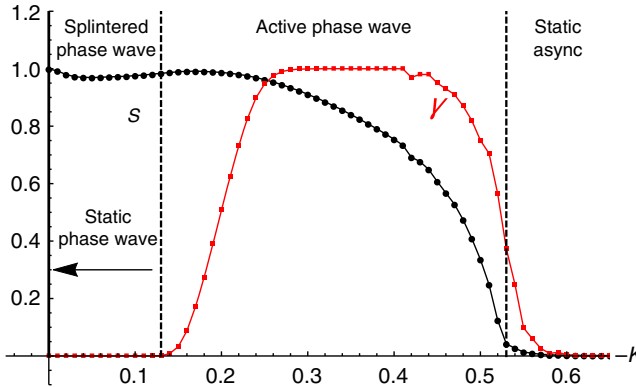

**Fig. 6** Order parameters. Asymptotic behavior of the order parameter $S := \max(S_+, S_-)$ (black dots) and $\gamma$ (red dots) for $J = 0.5$ and $N = 800$. Note the bifurcation of $S$ from at $K \approx -0.53$ near the approximation (18) $K_c = -1.2J = -0.6$. Data were collected using Heun's method for $T = 1000$ time units with stepsize $dt = 0.01$, of which the first half were discarded as transients. Each data point represents the average of one hundred realizations. Swarmalators were initially placed in a box of length 2 uniformly at random for all values of $K$ with a common seed, while their phases we drawn from $[-\pi, \pi]$

stationary splintered and active phase wave states, and is one in the static phase wave state.

Notice however that since $S_\pm$ is non-zero for both the splintered and active phase wave, it cannot distinguish between these states. To do this, we use another order parameter $\gamma$. We define this to be the fraction of swarmalators that have executed at least one full cycle in phase and position, after transients have been discarded. Then $\gamma$ is zero for the splintered phase wave, and non-zero for the active phase wave. Using $\gamma$ in concert with $S_\pm$ then allows us to discern all the macroscopic states of our system as illustrated in Fig. 6.

**Stability analysis**. To calculate the critical coupling strength $K_c$ at which the static async state loses stability, we consider perturbations $\eta$ in density space defined by

$$\rho(\mathbf{x}, \theta, t) = \rho_0(\mathbf{x}, \theta) + \varepsilon\eta(\mathbf{x}, \theta, t) \quad (13)$$

where $\rho_0(\mathbf{x}, \theta, t) = (4\pi^2)^{-1}g_1(r)$ is density in the static async state. In Supplementary Note 2, we substitute this ansatz into the

continuity equation, expand $\eta$ in a Fourier series, $\eta(\mathbf{x}, \theta, t) = \sum_{n=0} b_n(\mathbf{x}, t)e^{in\theta} + c.c.$, and derive evolution equations for the harmonics $b_n$. We show the critical mode is $b_1(\mathbf{x}, t)$ (higher modes are zero, and the zeroth mode is stable) which obeys

$$
\dot{b}_1(\mathbf{x}, t) = -\frac{J}{2}\nabla\rho_0(\mathbf{x}) \cdot \int \frac{\tilde{\mathbf{x}} - \mathbf{x}}{|\tilde{\mathbf{x}} - \mathbf{x}|} b_1(\tilde{\mathbf{x}}, t) \, d\tilde{\mathbf{x}} \\
+ \frac{(J + K)}{2}\rho_0(\mathbf{x})\int \frac{1}{|\tilde{\mathbf{x}} - \mathbf{x}|} b_1(\tilde{\mathbf{x}}, t) d\tilde{\mathbf{x}}.
$$

$$(14)$$

We next expand in an additional Fourier series: $b_1(r, \phi, t) = \sum_{m=0}^{\infty} f_m(r, t)e^{im\phi} + c.c.$. Substituting this ansatz into (14) leads to a evolution equation for each mode $f_m(r, t)$. We then set $f_m(r, t) = e^{\lambda_m t}c_m(r)$ and derive the following eigenvalue equation:

$$\lambda_m c_m(r) = \int_0^R H_m(r, s)c_m(s)s\,ds \quad (15)$$

where $R$ is the radius of the support of the density in the static async state. We focus first on the zeroth mode $f_0$ for which we can compute $H_0(r, s)$ analytically:

$$\lambda_0 c_0(r) = \int_0^R H_0(r, s)c_0(s)s\,ds, \quad (16)$$

$$
H_0(r, s) = \frac{J(r^2 - s^2)g'(r) + 2rg(r)(J + K)}{4\pi^2 r(r + s)}\mathcal{K}\left(\frac{4rs}{(r + s)^2}\right) \\
+ \frac{J(r + s)g'(r)}{4\pi^2 r}\mathcal{E}\left(\frac{4rs}{(r + s)^2}\right)
$$

$$(17)$$

where $\mathcal{K}, \mathcal{E}$ are the complete elliptic integral of the first and second kinds. We were unable to solve (16) for $\lambda_0$ analytically. Instead, we found it numerically by approximating the integral using Gaussian quadrature. This reduces (16) to the form $\lambda' c_i = M_{ij}c_j$ where $M_{ij} = H_0(r_i, r_j)w_j$, $w_j$ are Gaussian quadrature weights, $r_i = i * (R/N')$ and $i = 1 \ldots N'$. The eigenvalues $\lambda'_0(N')$ of $M_{ij}$, which depend on the number of grid points $N'$ used in the quadrature, then approximate $\lambda_0$.

The eigenvalues $\lambda'_0(N')$ have unexpected properties. The real part of the most unstable eigenvalue, denoted $\lambda_0^{*'}(N')$, is positive for all $J, K$. This tells us that $f_0$ is always unstable, which in turn tells us that the static async state is always unstable!

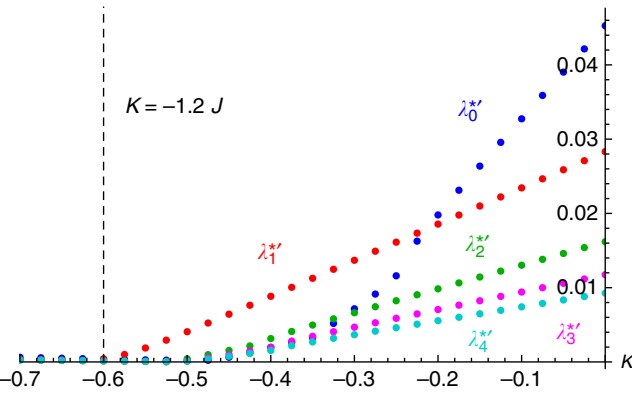

**Fig. 7** Spectrum. The real part of the most unstable eigenvalue, $\lambda_m^{*'}$, of the first five modes $f_m$ calculated from Eq. (15) for $J = 0.5$. Notice that they are all is positive for all $K$. Each $\lambda_m^{*'}$ was calculated by approximating the integral of the R.H.S. of (15) using Gaussian quadrature with $N' = 200$ grid points and diagonalizing the resulting matrix. The upper limit of integration $R = 1.15$ was measured from simulations. The radial density $g(r)$ was determined numerically as discussed in Supplementary Note 1. The kernels $H_m$ in Eq. (15) for $m > 1$ were calculated numerically. The dashed line marks the approximation to the critical coupling strength (18)

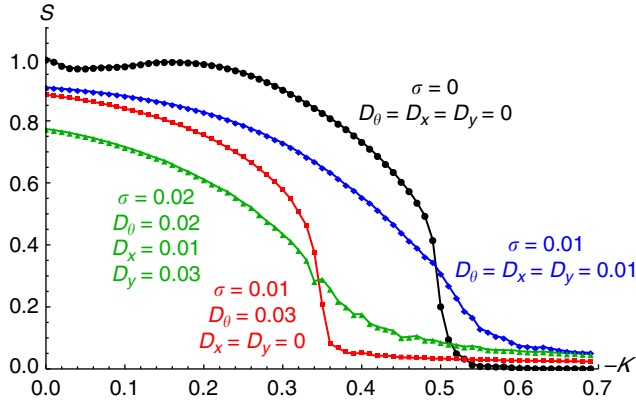

**Fig. 8** Order parameter with disorder. Asymptotic behavior of order parameter $S = \max(S_+, S_-)$ vs. $K$ for $J = 0.5$ and different amounts of disorder as quantified by the width of the distribution of natural frequencies $\sigma$ and the noise strengths, $D_\theta$, $D_x$, and $D_y$. As can be seen, greater amount of disorder stabilize the async state, as indicated by $-K_c$ becoming smaller and smaller. Note also the disappearance of the dip in the $S(K)$ curve, which tells us the splintered phase wave state does not exist in the presence of noise of this strength. Simulations were run for $N = 500$ swarmalators using Heun's method for $T = 1000$ time units with stepsize $dt = 0.01$, the first half of which were discarded. Each data point represents the average of 10 realizations

In Fig. 7 we plot $\lambda_0^{*'}(N')$ vs. $K$ for $J = 0.5$ and $N' = 200$ grid points. As can be seen it is small but positive for sufficiently negative $K$. Note however that there is a transition-like point $K_0^* \approx -0.5$ beyond which $\lambda_0^{*'}(N')$ increases sharply. Figure 7 also shows $\lambda_m^{*'}(N')$ for $m = 1, 2, 3, 4$, which have the same behavior as $\lambda_0^{*'}(N')$: they are small but positive for $K < K_m^*$, and grow sharply for $K > K_m^*$.

Small but positive eigenvalues for $K < K_m^*$ were a surprise. We were expecting them to be negative, since simulations show the static async state is stable. We were thus suspicious of these results, and doubted the accuracy of the approximation $\lambda_0^{*'}(N')$ to the true $\lambda_0^*$. We therefore repeated the calculation for different values of $N'$ up to $N' = 1600$ in Supplementary Note 2. Contrary to our expectations, we found that while the $\lambda_m^{*'}(N')$ got smaller, they consistently remained positive for $K < K_m^*$.

We also crudely investigated the $N' \to \infty$ limit in two ways: first, by fitting our data to curves of the form $a + b(N')^c$ and second, by using Richardson extrapolation. Due to the small magnitudes of the $\lambda_m^{*'}(N')$ however, the results were rather unconvincing. Typical values for the best fit parameter $a$, which represents the limiting behavior of $\lambda_m^*$, were $a \sim 10^{-6}$. The confidence interval for this parameter also contained positive and negative values. On top of that, the approximations from the first and second methods gave inconsistent results. Hence we were unable to reliably determine the sign of $\lambda_m^{*'}$ when $K < K_m^*$ and $N' \to \infty$, which preventing us from accurately ascertaining the stability of the static async state. We restate however that the fact that $\lambda_m^{*'} > 0$ for the large but finite value of $N'$ we used is significant evidence that the unanticipated instability of the static async state is genuine.

While a rigorous determination of the sign of $\lambda_m^{*'}$ when $K < K_m^*$ remains elusive, our analysis certifiably shows its magnitude is very small. Hence, whatever the stability or instability of the $m$-th mode $f_m$ turns out to be, it must be weak. In turn, then, the static async state has weak stability properties for $K < K_c$, where $K_c = \min_m K_m^*$ (i.e., at the point the most unstable $f_m$ loses stability). To find this $K_c$ we look for the least

stable mode. In Fig. 7 we see the $f_1$ becomes unstable first. There are of course an infinite number of modes, but as can be seen, $\lambda_m^*$ appears to decrease with increasing $m$. Thus we assume $\min_m K_m^* = 1$. In Supplementary Note 2, we approximate $K_1^* = \mathrm{argmax} \frac{d^2 \lambda_1^*}{dK^2}$, calculate it for different $J$, and find the following linear relation:

$$K_c \approx -1.2J. \tag{18}$$

Summarizing our main result: in the continuum limit $N \to \infty$, the static async state is unstable for $K > K_c$, and either weakly stable, neutrally stable, or weakly unstable for $K < K_c$. Further, numerical evidence suggests that the third option, weakly unstable, is the most likely. While this result is perhaps unsatisfying from a technical perspective, in practice it has utility. For example as shown in Fig. 1, the approximation (18) for $K_c$ agrees reasonably well with finite $N$ simulations.

**Genericity**. Our analysis so far has been for the instance (3), (4), of the model defined by (1), (2). This begs the question of whether the phenomena we found are generic to the model, or specific to this instance of the model. To answer this question, we ran simulations for different choices of the functions $\mathbf{I}_{rep}$, $\mathbf{I}_{att}$ and $G$; see Supplementary Note 4.

In all but one case, we found the same phenomena. The exception is when a linear attraction kernel $\mathbf{I}_{att}(\mathbf{x}) = \mathbf{x}$ is used. Here we found new states, which we call 'non-stationary phase waves'. They are similar to the active phase wave, except now the phase $\Psi_\pm$ of the order parameter $W_\pm$ begins to rotate, reminiscent of the traveling wave states found in the Kuramoto model with distributed coupling strengths[53,54]. We further discuss this and other properties in Supplementary Note 4.

**Noise and disordered natural frequencies**. The swarmalators previously considered were identical and noiseless. We now relax

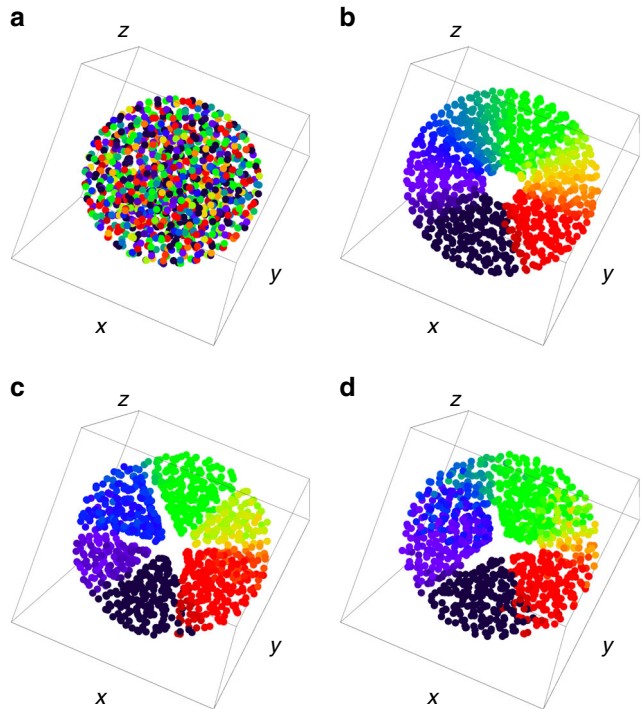

**Fig. 9** Scatter plots in 3D. Four states in the $(x, y, z)$ plane, where the swarmalators are colored according to their phase. Data were collected for $J = 0.5$ and $N = 1000$ swarmalators for $T = 5000$ time units with stepsize $dt = 0.001$ using Heun's method. **a** Static async state for $K = -1$. **b** Static phase wave for $K = 0$ **c** Splintered phase wave for $K = -0.05$. **d** Active phase wave state for $K = -0.6$. Supplementary Movies 9–12 correspond to panels **a**–**d**

these idealizations. Then the governing equations are

$$\dot{\mathbf{x}}_i = \frac{1}{N} \sum_{j \neq i}^{N} \left[ \frac{\mathbf{x}_j - \mathbf{x}_i}{|\mathbf{x}_j - \mathbf{x}_i|} \left( 1 + J \cos(\theta_j - \theta_i) \right) - \frac{\mathbf{x}_j - \mathbf{x}_i}{|\mathbf{x}_j - \mathbf{x}_i|^2} \right] + \xi_i^{\mathbf{x}}(t),$$
(19)

$$\dot{\theta}_i = \omega_i + \frac{K}{N} \sum_{j \neq i}^{N} \frac{\sin(\theta_j - \theta_i)}{|\mathbf{x}_j - \mathbf{x}_i|} + \eta_i(t),$$
(20)

where $\omega_i$ are random variables drawn from a Lorentzian $g(\omega) = (\sigma/\pi) \left[ (\omega - \mu)^2 + \sigma^2 \right]^{-1}$. By a change of frame we set $\mu = 0$, leaving just $\sigma$, which quantifies the strength of the disorder. We choose white noise variables $\eta_i(t)$ and $\xi_i^x(t)$ with zero mean and strengths $D_x$, $D_y$, $D_\theta$ characterized by $\langle \xi_i^x(t) \xi_j^x(t') \rangle = 2D_x \delta_{ij} \delta(t - t')$, etc.

Simulations show that when just phase noise $D_\theta$ is turned on, noisy versions of all the states are realized. The splintered phase wave however degenerates into the active phase wave for all but the smallest noise $D_\theta \gtrsim 10^{-3}$. In the remaining states, the spatial densities remain compact supported with the same radii, except now the swarmalators have noisy phase motion (this induces some spatial movement, which disappears when $N \to \infty$ as we show in Supplementary Note 3). Hence the following states, where we have swapped the descriptor 'static' with 'noisy', are robustly realized when $D_\theta > 0$: noisy phase wave, active phase wave, noisy async.

Frequency disorder $\sigma > 0$ has a more serious effect. Since $g(\omega)$ is symmetric about zero, there are equal numbers of swarmalators with oppositely signed natural frequencies. This turns the static/noisy phase wave into the active phase wave, in

the sense that counter-rotating groups develop. This is not seen in the async state. Here, there are noisy spatial movements which vanish as $N \to \infty$, as in the noisy async state. In contrast however, the swarmalators execute noisy, but full, phase cycles. To highlight this distinction, we rename the state the active async state. The states realized are then the active phase wave and the active async.

Finally spatial noise $D_x$, $D_y > 0$ simply blurs the spatial densities of the states. No other phenomena are induced. Hence when $D_\theta$, $\sigma$, $D_x$, $D_y > 0$, we again get the active phase wave and active async states.

In Fig. 8 we plot the order parameter $S(K)$ for different amounts of noise and frequency disorder. As for the original model, $S$ simply declines to zero as $K$ is decreased, with the noise and disordered frequencies changing just the shape of the curves and the value of $K_c$. Note the disappearance of the dip in $S$ for small $K$, which indicates the absence of the splintered phase wave state. Note also we do not plot the second order parameter $\gamma$ which discerns the splintered phase wave since this state does not robustly exist when $\sigma$, $D_\theta \neq 0$.

**Swarmalators in 3D.** So far we have considered swarmalators moving in two dimensions. While there are physical systems where this approximation is valid, such as certain active colloids[55] or sperm, which are often attracted to two-dimensional surfaces[56], this restriction was mostly for mathematical convenience. Here we explore the more physically realistic case of motion in three spatial dimensions (in Supplementary Note 6 we also explore motion in one dimension). For simplicity we consider the case of identical swarmalators with no noise, although we relax these idealizations in Supplementary Note 5. Our system is then

$$\dot{\mathbf{x}}_i = \frac{1}{N} \sum_{j \neq i}^{N} \left[ \frac{\mathbf{x}_j - \mathbf{x}_i}{|\mathbf{x}_j - \mathbf{x}_i|} \left( 1 + J \cos(\theta_j - \theta_i) \right) - \frac{\mathbf{x}_j - \mathbf{x}_i}{|\mathbf{x}_j - \mathbf{x}_i|^3} \right],$$
(21)

$$\dot{\theta}_i = \frac{K}{N} \sum_{j \neq i}^{N} \frac{\sin(\theta_j - \theta_i)}{|\mathbf{x}_j - \mathbf{x}_i|},$$
(22)

where $\mathbf{x}_i = (x_i, y_i, z_i)$. These are the same as Eqs. (3) and (4), except the exponent of the hard shell repulsion is now 3 (we choose this because it yields simple formulas for the radii of certain states).

Simulations show that analogs of the states found in 2D are realized. We show these as scatter plots in the $(x, y, z)$ plane in Fig. 9. We also provide movies of the evolution to these states in Supplementary Movies 9–12. The static sync and async states become spheres (note we do not plot the static sync state due to space limitations) as seen in panel (a). As in the 2D case, we can calculate their radii when a linear attraction kernel is used,

$$R_{\text{sync}} = (1 + J)^{-1/3},$$
(23)

$$R_{\text{async}} = 1,$$
(24)

which agree with simulation as shown in Supplementary Note 5.

In panel (b) we show the static phase wave becomes a sphere with a cylindrical hole through its center. The orientation of this cylinder is determined by the initial conditions. The phase and azimuthal angle $\phi = \tan^{-1}(y/x)$ are correlated in the same way for each value of the polar angle $\alpha = \cos^{-1}\left(z/\sqrt{x^2 + y^2 + z^2}\right)$ (when the azimuthal and polar angles are measured relative to the axis of the cylindrical hole). We show this more clearly in a scatter plot in the $(\theta, \phi)$ plane in Supplementary Note 5.

As in the 2D model, this correlation between $\phi$ and $\theta$ persists for the splintered phase waves and active phase wave states as can be seen in panels (c) and (d) of Fig. 9. The motion of the swarmalators in these states are as before: in the splintered phase wave they 'quiver', executing small oscillations in space and phase, while in the active phase wave they execute full rotations (note the spatial component of these rotations are in the azimuthal direction $\hat{\phi}$ only, not in the polar direction $\hat{\alpha}$). In Supplementary Note 5 we show how the order parameters $S_{\pm}$, $\gamma$ can also be used to differentiate these 3D states.

**Alignment and self-propulsion.** Up to now we have considered the trivial case of swarmalators that propel themselves with constant magnitude and direction, in a manner uninfluenced by their neighbors. This allowed us to set this term to zero via a change of reference. In many real systems, however, such behavior is unrealistic: individuals often adjust the direction of their motion to align with that of their neighbors. Vicsek studied this alignment effect in a seminal work[29].

We here partially explore the effect of alignment on swarmalator systems. Accordingly we endow each swarmalator with an orientation $\beta$, which characterizes the direction of its self-propulsion. The inclusion of alignment makes our model complicated; there are now four state variables $(x, y, \theta, \beta)$ per swarmalator, which could interact with each other in potentially many ways. Furthermore, there are six parameters $(J, K, \sigma, D_{\theta}, D_x, D_y)$, not to mention any additional parameters governing the evolution of $\beta$. An exhaustive study of orientable swarmalators is thus beyond the scope of the present work. Hence, we restrict ourselves to checking if the states of our swarmalator system are robust to the inclusion of simple alignment dynamics.

To this end, we study the simplest possible extension to our current model: we choose Vicsek type interactions between $\mathbf{x}$ and $\beta$, and leave $\beta$ and the phase $\theta$ uncoupled (although they are indirectly coupled through the position $\mathbf{x}$). Our system then reads

$$\dot{\mathbf{x}}_i = \frac{1}{N} \sum_{j \neq i}^{N} \left[ \frac{\mathbf{x}_j - \mathbf{x}_i}{|\mathbf{x}_j - \mathbf{x}_i|} \left( 1 + J\cos(\theta_j - \theta_i) \right) - \frac{\mathbf{x}_j - \mathbf{x}_i}{|\mathbf{x}_j - \mathbf{x}_i|^2} \right] \quad (25)$$
$$+ \xi_i^{\mathbf{x}}(t) + v_0 \hat{n},$$

$$\dot{\theta}_i = \omega_i + \frac{K}{N} \sum_{j \neq i}^{N} \frac{\sin(\theta_j - \theta_i)}{|\mathbf{x}_j - \mathbf{x}_i|} + \eta_i(t), \quad (26)$$

$$\dot{\beta}_i = -\beta_i + \frac{1}{|\Lambda_i|} \sum_{j \in \Lambda_i} \beta_j + \zeta_i(t), \quad (27)$$

where $\hat{n} = (\cos\beta, \sin\beta)$, $\Lambda_i$ is the set of swarmalators within a distance $\delta$ of the $i$-th swarmalator, and $|\Lambda_i|$ is the number of such neighbors. The $\zeta_i(t)$ is a white noise variable with zero mean and strength $D_{\beta}$ characterized by $\langle \zeta_i(t)\zeta_j(t') \rangle = 2D_{\beta}\delta_{ij}\delta(t - t')$.

Simulations show that for certain parameter values aligned versions of all our states persist. We plot two of these in panels (a) and (b) of Fig. 10, where each swarmalator is depicted as a colored arrow, oriented according to $\beta$, and colored according to phase. As can be seen the swarmalators are aligned, with their space-phase distributions being the same as before. In contrast to the original model, however, the center of mass of each distribution now moves (in a direction determined by the initial conditions). In this sense the states are mobile. They are however equivalent to their static versions via a change of reference frame, $\mathbf{x} \rightarrow \mathbf{x} + \mathbf{v}_0 t$. For larger $D_{\beta}$, the unaligned versions of the same states are realized, as illustrated in panels (c) and (d) of Fig. 10.

We have demonstrated that the phenomena of our system are insensitive to the inclusion of simple alignment dynamics. We restate however that we have not comprehensively explored the space defined by the other parameters $(J, K, \sigma, v_0, D_x, D_y)$ given its large size. Thus it remains to be seen if new states will be found.

**Discussion**

We have examined the collective dynamics of swarmalators. These are mobile particles or agents with both phase and spatial degrees of freedom, which lets them sync and swarm. Furthermore, their phase and spatial dynamics are coupled. By studying simple models, we found this coupling leads to rich spatio-temporal patterns which we explored analytically and numerically. These patterns were robust to modifications to the model, namely motion in one, two, and three spatial dimensions, distributed natural frequencies, noisy interactions, and alignment dynamics. We thus believe they could be realized in nature or technology.

A pertinent future goal, then, is to investigate the behavior of real-world systems of swarmalators. As mentioned in the introduction, colloidal suspensions of magnetic particles[42–44] or active spinners[45,46] are promising candidates. For example, structures equivalent to the static phase wave state have been experimentally realized by Snezhko and Aranson, when studying the behavior of ferromagnetic colloids at liquid-liquid interfaces[43] (the particles comprising the colloids can be considered swarmalators if we interpret the angle subtended by their magnetic dipole vectors as their phase). As shown in Fig. 4 of ref. [43], the colloids can form asters. These are structures composed of radial chains of magnetically ordered particles, which "decorate slopes of a self-induced circular standing wave"[43], analogous to the annular pattern of correlated phases and positions of the static phase wave shown in Fig. 2c.

Perhaps colloidal equivalents of the splintered and active phase wave states could also be realized. Aside from being theoretically interesting, the ability to engineer these states could have practical application. For instance, Snezhko and Aranson also show that asters can be manipulated to capture and transport target particles. The non-stationary behavior of the splintered and active wave states might also have locomotive utility. Tentative evidence for this claim is provided by populations of cilia, whose collective metachronal waves, similar to the motion of swarmalators in the aforementioned states, are known to facilitate biological transport[57–59].

Other plausible systems of real-world swarmalators are biological microswimmers, self-propelled micro-organisms capable of collective behavior[60]. One such contender is populations of spermatoza, which exhibit rich swarming behavior such as trains[61,62] and vortex arrays[52], the latter of which is reminiscent of the active phase wave state, as mentioned in the Results section. The phase variable for each sperm is associated with the rhythmic beating of the sperm's tail, which can synchronize with that of a neighboring sperm[63,64]. It has been theorized that this can induce spatial attraction[65], leading to clusters of synchronized sperm, consistent with experimentally observed behavior[66].

There are also theoretical avenues to explore within our proposed model of swarmalators. For instance the curious stability properties of the static async state deserve further study. Another route would be to include more realism by including heterogeneity in the coupling parameters $K$, $J$, or by choosing more complicated interaction functions $\mathbf{I}_{att}$, $\mathbf{I}_{rep}$, $G$, $H_{att}$. For example we chose $H_{att}(\theta) = \sin(\theta)$ to mimic the Kuramoto model, but as we saw, it led to just the trivial static sync state when $K > 0$. Perhaps choosing the more realistic Winfree model for the phase dynamics, which gives rise to richer collective behavior, would

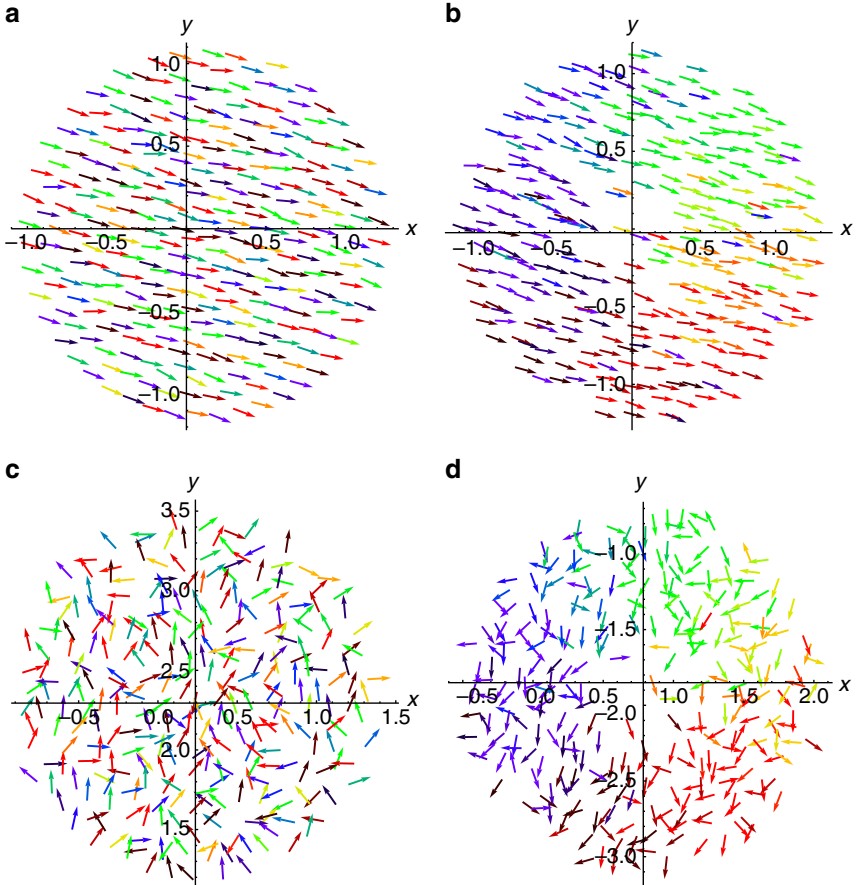

**Fig. 10** Scatter plots with alignment. Four states in the $(x, y)$ plane where the swarmalators are depicted as colored arrows, whose orientation represents $\beta$, and whose color represents the phase $\theta$. Data were collected for $N = 300$ swarmalators for $T = 5000$ time units with stepsize $dt = 0.01$ using Heun's method. In each panel, parameter values were $J = \delta = 0.5$, $\sigma = D_\theta = D_\beta = 0.01$, $D_x = D_y = 0$ and $v_0 = 0.001$. **a** Aligned active async for $(K, D_\beta) = (-1.0, 0.01)$. **b** Aligned noisy phase wave for $(K, D_\beta) = (-0.1, 0.01)$. **c** Unaligned active async for $(K, D_\beta) = (-1.0, 1.0)$. **d** Unaligned noisy phase wave for $(K, D_\beta) = (-0.1, 1.0)$

lead to more interesting swarmalator phenomena in this parameter regime.

Perhaps the most important direction for future work is to more fully explore the interplay among aggregation, alignment, and synchronization—or put another way, to explore the collective behavior of particles with a position $\mathbf{x}$, an orientation $\beta$, and an internal phase $\theta$. The primary goal of our work is to draw attention to this class of problems, which we believe define a wide landscape of emergent behavior. In this work, we have started to map out this landscape by studying a simple model that contains a subset of these three effects, namely aggregation and synchronization.

Others have considered the remaining subsets. For example, Leon and Liverpool have explored the interaction between alignment and synchronization[67]. They introduced a new class of soft active fluids whose units have an orientation and phase. They found this mixture can either enhance or inhibit the transition from disordered states to states with polar order. The latter states are roughly similar to the aligned static async states. They also found transitions from disordered states to states with phase order, which are analogous to unaligned static sync states. Yet counterparts of the static, splintered, and active phase waves were not reported.

The final combination, aggregation and alignment, is perhaps the most well studied, in both new models and old. For instance, Starnini et al.[68] recently introduced a model of mobile particles capable of aggregating and aligning their opinions, and found the emergence of echo chambers. Even in the classic Vicsek model

and its numerous extensions, new phenomena are still being found. For instance, Kruk et al.[69] found that delayed alignment in the Vicsek model produces self-propelled chimeras; perhaps delayed phase interactions could lead to similar states for swarmalators. Liebchen and Levis[70] considered units with an intrinsic rotation, and found 'phase separated droplets': clusters of rotation-synchronized particles surrounded by a sea of incoherent particles (multiple droplets are also possible). These droplets are similar to our static sync states, but they differ in the crucial respect that the entire population is synchronized in our static sync state. Here too, the counterparts of our static, splintered, and active phase waves were not seen.

Thus, to the best of our knowledge, no other models display states analogous to the splintered phase waves and active phase waves found in our swarmalator model. In that sense, those two states are unprecedented.

## Methods

**Properties of static sync and async state**. We here use techniques used by Fetecau et al.[50] when studying swarm dynamics to study the static sync and static async states with a linear attraction kernel $\mathbf{I}_{att}(\mathbf{x}) = \mathbf{x}$ is used. We start with the async state whose density is

$$\rho(r, \phi, \theta, t) = \frac{1}{4\pi^2} g_1(r), \quad 0 \leq r \leq R. \tag{28}$$

We wish to solve for the radial density $g_1(r)$ and the radius $R$ of its support. In this state the swarmalators are at rest and their phases are unchanging, so $\underline{v} \equiv \underline{0}$, where $\underline{v} = (v_x, v_y, v_\theta)$. As we will show, it is also useful to consider the divergence of the velocity, which must also be zero (from the continuity equation for the conservation of swarmalators, and by applying the assumptions that the density for the

static async state is stationary and the velocity is zero). This gives us a pair of simultaneous equations,

$$\underline{v} \equiv \underline{0}, \tag{29}$$

$$\nabla \cdot \underline{v} \equiv 0. \tag{30}$$

We begin with divergence term given by (30). In Cartesian coordinates the velocity reads

$$\mathbf{v_x}(\mathbf{x}, \theta, t) = \int \left( (\tilde{\mathbf{x}} - \mathbf{x})(1 + J\cos(\tilde{\theta} - \theta)) - \frac{\tilde{\mathbf{x}} - \mathbf{x}}{|\tilde{\mathbf{x}} - \mathbf{x}|^2} \right) \rho(\tilde{\mathbf{x}}, \tilde{\theta}, t) \mathrm{d}\tilde{\mathbf{x}} \, \mathrm{d}\tilde{\theta}, \tag{31}$$

$$v_\theta(\mathbf{x}, \theta, t) = \int \frac{\sin(\tilde{\theta} - \theta)}{|\tilde{\mathbf{x}} - \mathbf{x}|} \rho(\tilde{\mathbf{x}}, \tilde{\theta}, t) \mathrm{d}\tilde{\mathbf{x}} \, \mathrm{d}\tilde{\theta}. \tag{32}$$

The divergence has a spatial and phase component: $\nabla \cdot \underline{v} = \nabla_{\mathbf{x}} \cdot \mathbf{v_x} + \partial_\theta v_\theta$. The phase component $\partial_\theta v_\theta$ is trivially zero, since the swarmalators' phases are uniformly distributed in phase in the static async state. We find the spatial component by applying $\nabla_{\mathbf{x}}$ to (31):

$$\nabla_{\mathbf{x}} \cdot \mathbf{v_x} = \int -2(1 + J\cos(\tilde{\theta} - \theta)) \rho(\tilde{\mathbf{x}}, \tilde{\theta}, t) \mathrm{d}\tilde{\mathbf{x}} \mathrm{d}\tilde{\theta} \tag{33}$$

$$+ 2\pi\delta(\tilde{\mathbf{x}} - \mathbf{x}) \rho(\tilde{\mathbf{x}}, \tilde{\theta}, t) \, \mathrm{d}\tilde{\mathbf{x}} \mathrm{d}\tilde{\theta}. \tag{34}$$

Here we have used the identity (expressed most cleanly in Cartesian coordinates)

$$\nabla_{\mathbf{x}} \cdot \frac{\tilde{\mathbf{x}} - \mathbf{x}}{|\tilde{\mathbf{x}} - \mathbf{x}|^2} = -2\pi\delta(\tilde{\mathbf{x}} - \mathbf{x}). \tag{35}$$

Simplifying this, and substituting $\partial_\theta v_\theta = 0$ gives the full divergence

$$\nabla \cdot \underline{v} = 2\pi\rho(\mathbf{x}, \theta) - 2\int (1 + J\cos(\tilde{\theta} - \theta)) \rho(\tilde{\mathbf{x}}, \tilde{\theta}, t) \mathrm{d}\tilde{\mathbf{x}} \, \mathrm{d}\tilde{\theta}. \tag{36}$$

By (30) we require this to be zero, which gives a self-consistent equation for $\rho$:

$$\rho(\mathbf{x}, \theta, t) = \frac{1}{\pi} \int (1 + J\cos(\tilde{\theta} - \theta)) \rho(\tilde{\mathbf{x}}, \tilde{\theta}, t) \mathrm{d}\tilde{\mathbf{x}} \, \mathrm{d}\tilde{\theta}. \tag{37}$$

Finally substituting the ansatz given by (28) into this and performing the integration over $\phi$ gives

$$g_1(r) = 2\int_0^R g_1(\tilde{r}) \tilde{r} \, \mathrm{d}\tilde{r} = M = \mathrm{const}. \tag{38}$$

This tells us $\rho$ is constant inside a disc of radius $R$. The radius $R$ can be determined via self-consistency: $M = \int_0^R rg(r)\mathrm{d}r = \int_0^R rM\mathrm{d}r \Rightarrow R = 1$. By normalizing $\rho$ as per (28) we find $M = 1$ which means $g(r) = 2$. Putting this all together gives

$$\rho_{\mathrm{async}}(r, \phi, \theta, t) = \frac{1}{2\pi^2}, \quad 0 \le r \le R_{\mathrm{async}} \tag{39}$$

$$R_{\mathrm{async}} = 1. \tag{40}$$

We must now check if the solutions given by (39) and (40) imply $\underline{v} \equiv \underline{0}$ as required by (29). We do this in Cartesian coordinates, in which

$$\rho(\mathbf{x}, \theta, t) = \frac{1}{\pi R^2} \delta(\theta - \theta_0), \quad |\mathbf{x}| \le R, \tag{41}$$

where $\theta_0$ is the final, common phase of each swarmalator. Substituting this into Eqs. (31) and (32) for $v_{\mathbf{x}}, v_\theta$ and performing the integration gives

$$\mathbf{v_x}(\mathbf{x}, \theta, t) = \frac{1}{4\pi R^2} (R^2 - [1 + J\cos(\theta - \theta_0)]) \mathbf{x}, \tag{42}$$

$$v_\theta(\mathbf{x}, \theta, t) = 0, \tag{43}$$

where we have used the identity

$$\int_{|\tilde{\mathbf{x}}| < R} \frac{\tilde{\mathbf{x}} - \mathbf{x}}{|\tilde{\mathbf{x}} - \mathbf{x}|^2} = \pi\mathbf{x}, \quad |\mathbf{x}| < R. \tag{44}$$

We see that $\mathbf{v_x} = 0$ at $\theta = \theta_0$ if $R = 1$, as required. Hence we have shown that the solutions (39), (40) satisfy Eqs. (29) and (30).

Carrying out the same analysis for the static sync state leads to

$$\rho_{\mathrm{sync}}(r, \phi, \theta, t) = \frac{1}{\pi^2} \delta(\theta - \theta_0), \quad 0 \le r \le R_{\mathrm{sync}} \tag{45}$$

$$R_{\mathrm{sync}} = (1 + J)^{-1/2}, \tag{46}$$

where $\theta_0$ is the final common phase of the swarmalators in the static sync state.

**Properties of static phase wave state.** Here we calculate the density of swarmalators, and inner and outer radii $R_1$, $R_2$ of the annulus, in the static phase wave state, when a linear attraction kernel is used.

The calculation is the same as for the static sync and async states: we assert

$$\underline{v} \equiv \underline{0}, \tag{47}$$

$$\nabla \cdot \underline{v} \equiv 0. \tag{48}$$

The density of the static phase wave state is

$$\rho(r, \phi, \theta, t) = (2\pi)^{-1} g_2(r) \delta(\phi - \theta), \quad R_1 < r < R_2. \tag{49}$$

We first calculate the divergence, which in polar coordinates is given by

$$\nabla \cdot \underline{v} = \frac{1}{r} \frac{\partial (rv_r)}{\partial r} + \frac{\partial (rv_\phi)}{\partial \phi} + \frac{\partial v_\theta}{\partial \theta}. \tag{50}$$

The velocity $\underline{v} = (v_r, v_\phi, v_\theta)$ is given by

$$v_r = \int (\tilde{r}\cos(\tilde{\phi} - \phi) - r)(1 + J\cos(\tilde{\theta} - \theta) \\ - \frac{1}{\tilde{r}^2 - 2r\tilde{r}\cos(\tilde{\phi} - \phi) + r^2}) \rho(\tilde{r}, \tilde{\phi}, \tilde{\theta}) \tilde{r} \, \mathrm{d}\tilde{r} \, \mathrm{d}\tilde{\phi} \, \mathrm{d}\tilde{\theta}, \tag{51}$$

$$v_\phi = \int \tilde{r}\sin(\tilde{\phi} - \phi)(1 + J\cos(\tilde{\theta} - \theta) \\ - \frac{1}{\tilde{r}^2 - 2r\tilde{r}\cos(\tilde{\phi} - \phi) + r^2}) \rho(\tilde{r}, \tilde{\phi}, \tilde{\theta}) \tilde{r} \, \mathrm{d}\tilde{r} \, \mathrm{d}\tilde{\phi} \, \mathrm{d}\tilde{\theta}, \tag{52}$$

$$v_\theta = \int \frac{\sin(\tilde{\theta} - \theta)}{\tilde{r}^2 - 2r\tilde{r}\cos(\tilde{\phi} - \phi) + r^2} \rho(\tilde{r}, \tilde{\phi}, \tilde{\theta}) \tilde{r} \, \mathrm{d}\tilde{r} \, \mathrm{d}\tilde{\phi} \, \mathrm{d}\tilde{\theta}. \tag{53}$$

Taking the derivatives on these, plugging in Eq. (49) for $\rho$, and substituting the result into (50), gives

$$\underline{\nabla} \cdot \underline{v} = -2 + g_2(r) + \frac{J}{2r} \int_{R_1}^{R_2} \tilde{r}^2 g_2(\tilde{r}) \mathrm{d}\tilde{r}. \tag{54}$$

Setting this to zero, we see $g_2(r)$ satisfies

$$g_2(r) = 2 - \frac{J}{2r} \int_{R_1}^{R_2} \tilde{r}^2 g_2(\tilde{r}) \mathrm{d}\tilde{r}, \tag{55}$$

which means it can be determined self-consistently in terms of $R_1$ and $R_2$. The result is

$$g_2(r) = 1 - \frac{\Gamma_J}{r}, \quad R_1 \le r \le R_2 \tag{56}$$

with $\Gamma_J = 2J(R_2^3 - R_1^3)(3J(R_2^2 - R_1^2) + 12)^{-1}$.

Next we use the result (56) in $\underline{v} = \underline{0}$ to compute the inner and outer radii $R_1$, $R_2$. We first evaluate $v_r$ by substituting (56) into (51). Performing the integration we get

$$v_r(r) = Cr + \frac{D}{r} \tag{57}$$

with

$$C = -R_1^2 + \frac{4JR_1(R_2^3 - R_1^3)}{3J(R_2^2 - R_1^2) + 12}, \tag{58}$$

$$D = 1 + \frac{R_2 - R_1}{6} \left[ -6(R_2 - R_1) + \frac{8JR_1(R_2^3 - R_1^3)}{3J(R_2^2 - R_1^2) + 4} \right]. \tag{59}$$

Since $\underline{v} \equiv \underline{0}$, the coefficients $C, D$ must be zero. This yields two equations for $R_1, R_2$,

with solutions

$$R_1 = \Delta_J \frac{-\sqrt{3}J - 3\sqrt{12 - 5J}\sqrt{J+4} + 12\sqrt{3}}{12J}, \tag{60}$$

$$R_2 = \frac{\Delta_J}{2\sqrt{3}}, \tag{61}$$

with

$$\Delta_J = \sqrt{\frac{3J - \sqrt{36 - 15J}\sqrt{J+4} - 12}{J - 2}} \tag{62}$$

and small-$J$ expansion given by

$$R_1 = \frac{J}{3} + O(J^2), \tag{63}$$

$$R_2 = 1 + \frac{J}{6} + O(J^2). \tag{64}$$

**Data availability**. The data that support the findings of this study (simulation source code and figure raw data) are available from the author K.P.O.K. upon request.

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

## Acknowledgements

Research supported by United States NSF Grant Nos. DMS-1513179 and CCF-1522054 (S.H.S), and by South Korean NRF Grant No. NRF-2015R1D1A3A01016345 (H.H.).

## Author contributions

K.P.O.K. and S.H.S. conceived the research. K.P.O.K. and H.H. performed the numerics. K.P.O.K. performed the analysis and drafted the manuscript. All authors discussed the results, drew conclusions and edited the manuscript.

## Additional information

**Competing interests:** The authors declare no competing financial interests.

