## [Peer Review File · Nature Communications]

Reviewers' comments:

Reviewer #1 (Remarks to the Author):

In this paper, a new class of models is proposed to study the collective dynamics of moving units, each having a phase degree of freedom as well as a position vector. The authors report a variety of interesting behaviors exhibited by such systems found by simulation, some of which are explained theoretically.

I think this work might be worth publishing in Nature Communications, since it opens a new area of coupled oscillators and motivates researchers in the field of nonlinear dynamics and related fields to investigate the properties of such systems in detail. However, there are quite a few places to be corrected with the supplemental material and also the authors do not cite a pioneering work in this particular field. See below.

1. Similar models were proposed a decade ago by the late Dan Tanaka and studied by his group. Their relevant papers are as follows:

M. Iwasa, K. Iida, and D. Tanaka, Phys. Rev. E, vol. 81, 046220 (2010).

M. Iwasa and D. Tanaka, Phys. Rev. E, vol. 81, 066214 (2010).

M. Iwasa, K. Iida, and D. Tanaka, Phys. Rev. E, vol. 83, 036210 (2011).

D. Tanaka, Phys. Rev. Lett. vol. 99, 134103 (2007).

I advise the authors to read these papers and include them in the introduction of their manuscript.

2. Checking out the supplemental material, I found a lot of mistakes, errors, and inconsistencies. The authors might have submitted it without doing any careful examination. All of them must be exactly corrected.

Reviewer #2 (Remarks to the Author):

In the manuscript "Swarmalators: Oscillators that sync and swarm" the authors consider a simple generic model of oscillators with coupled phase and spatial dynamics, where the feedback of the oscillators' phase onto the spatial dynamics is considered in addition to the feedback of the location onto the Kuramoto phase dynamics. Such a model is potentially very interesting, since it might account for applications in biology, robotics, physics of magnetic colloids, and microfluidic active spinners and microswimmers.

As a case example, the authors study in detail a generalized Kuramoto model, extended by spatial kinematics of active particles under an attractive, phase dependent, and a repulsive potential. They demonstrate five types of collective self-organized states, depending upon the phase coupling parameter K and the position-phase coupling parameter J . The most interesting ones are two non-stationary states: splintered phase wave and active phase wave. The latter state has similarities to shear flow in some biological models.

In the proposed model, spatial attraction and repulsion between the elements, and the influence of spatial proximity on the phase dynamics, is described by power-law functions. The phase interaction is governed by a sine

function, as in the Kuramoto model. Such a choice allows the authors to provide analytically the stability analysis in the supplemental material.

In the parameter plane of spatial and phase interaction strengths, the regions of stability for five different types of patterns are presented. To distinguish different dynamical states, the authors use a Kuramoto-like order parameter, as well as some additional measure.

I believe that the spatio-temporal patterns resulting from this simple model of interacting mobile particles with internal dynamics are interesting for a broad community because they might have promising applications in real-world natural and technological systems. The paper is well written and well presented. Thus I think the paper could be suitable for Nature Communications.

However, some questions remain to be answered by the authors: How do their model and their results compare with the well-known Vicsek model and its extensions (e.g., Kruk, Maistrenko, Wenzel, and Koepl, arXiv1511.04738, and many other references) and with more sophisticated specific models available, e.g., for magnetic colloids and microswimmers? What are the differences of the present model? In how far does it go beyond that previous work? What is really new? A detailed discussion should be given.

Furthermore, throughout the manuscript, the "swarmalators" are free to move in the plane. Would it be possible to consider their motion in more realistic 3-dimensional space? How would this influence the analysis and variety of patterns?

Minor remarks:

The style and format seem to correspond to PRL rather than to Nature Communications.

All the calculations are shifted to the Supplemental Material, and the authors should check if some of these can be included

in the Methods Section of Nature Communications.

In the caption of fig.5 the time interval of simulation should be given.

Reviewer #3 (Remarks to the Author):

This paper is based on an interesting idea: what if synchronizing oscillators, as studied in various version of the Kuramoto model, also move in open space such that they tend to synchronize preferentially with neighboring oscillators, and their velocities are also influenced by their degree of synchronization with neighbors? Such a bidirectional coupling between a Kuramoto model and a collective motion model, is not an entirely new idea (see, e.g. models of so-called rippling waves in colonies of myxobacteria, as in Igoshin et al., PNAS 18, 14913 (2001)), but has never been studied before in any detail.

Unfortunately, in my opinion, the authors fail to deliver a nice exploration of this general problem. Their model is rather specific, and rather remote from both any real situation (I doubt the magnetic particles of Snezhko and Aranson have much to do with the current paper). The model is also far from standard models of collective motion (no self-propulsion, no alignment, no noise). I failed to see whether, on the synchronization part, the oscillators have distributed individual frequencies or not (I concluded that, most likely, the oscillators are identical, at odds with the Kuramoto model, but I am not sure!).

We are served a rather elementary numerical study presenting a catalog of 'phases' that are trivial enough to be amenable to a near-exact solution. The coupling functions are near global-coupling (especially for the attractive interaction), which probably explains the relative simplicity of the 'phases' observed.

On a more specific side, no attempt is made to study/evaluate finite-size effects, the nature of the transition between the 'phases', the robustness against noise, so that, in the end, the results remain rather limited in scope.

We thank the referees for their helpful comments, which have helped us improve the paper. Our reply is structured as follows. We first discuss general changes to the manuscript, and then respond to the referees' comments one-by-one.

General changes

1. When discussing the active phase wave state, we stated that the swarms executed shear flow. This was a mistake; shear flow is present in the (ϕ, θ) plane, where ϕ is the spatial angle and θ is the phase variable, but not in the (x, y) plane. In the latter case, there are also oscillations in the radial position. To correct this, we have updated the relevant paragraph (first paragraph in 'Active phase wave' subsection, p.5).
2. We have made a number of videos showing the evolution of the states of our system. We do this for both two and three spatial dimensions. These videos have been added to the Supplementary Information.
3. We discovered a mistake in our stability analysis of the static async state. In correcting it, we have found unexpected stability properties: the state appears to be weakly unstable in the $N \rightarrow \infty$ limit. We discuss this property in the main text ("Results" section, "Stability analysis" subsection) and Supplementary Information ("Stability of static async state" section, p.3 of Supplementary Information).
4. Included a section in the Supplementary Information on how to approximate the radial density $g(r)$ and radius R of the static async state when the unit vector interaction kernel $I_{att}(x) = x/|x|$ is used. ("Properties of the static sync and async states" section "Unit vector attraction kernel" subsection)
5. The paper now has three authors. Hyunsuk Hong was invited to join the original author team, because of her expertise in simulation and finite-size scaling. She performed extensive simulations and analysis that have helped us improve the paper in response to the referees' comments.

Referee 1

"the authors do not cite a pioneering work in this particular field --- I advise the authors to read these papers and include them in the introduction of their manuscript."

We thank the referee for bring the work of Tanaka and colleagues to our attention. We have read and cited the relevant papers, and discuss them in the Introduction. (See Introduction, second to last paragraph, beginning "Tanaka and colleagues...")

"Checking out the supplemental material, I found a lot of mistakes, errors, and inconsistencies. The authors might have submitted it without doing any careful examination. All of them must be exactly corrected."

We thank the referee for bring this to our attention, and have corrected the mistakes and inconsistencies.

Referee 2

"However, some questions remain to be answered by the authors: How do their model and their results compare with the well-known Vicsek model and its extensions (e.g., Kruk, Maistrenko, Wenzel, and Koepl, arXiv1511.04738, and many other references) and with more sophisticated specific models available, e.g., for magnetic colloids and microswimmers? What are the differences of the present model? In how far does it go beyond that previous work? What is really new? A detailed discussion should be given."

We agree that the relationship between our model and other models (i.e. the Vicsek model) was not clear in our first draft of the paper. We thank the referee for the advice to clarify these connections. We have added three expository paragraphs in the "Model" section, starting with "Before stating our results..." in which we detail specifically the relationship between our proposed model and previous models. We have also significantly expanded the Discussion section. In particular, the work by Vicsek et al. and Kruk et al. are discussed in the paragraph beginning "The final combination..." near the end of the Discussion.

"Furthermore, throughout the manuscript, the "swarmalators" are free to move in the plane. Would it be possible to consider their motion in more realistic 3-dimensional space? How would this influence the analysis and variety of patterns?"

Thanks for the suggestion. We have now explored the behavior of the model in 3D, and give our results in the new "Extensions to the model" section. As we discuss there, the variety of patterns found in the 2D model persist in 3D. As for analysis, we have computed the radii of the static sync and async states, and confirmed them with simulations.

"The style and format seem to correspond to PRL rather than to Nature Communications. All the calculations are shifted to the Supplemental Material, and the authors should check if some of these can be included in the Methods Section of Nature Communications"

We have moved a substantial portion of the stability analysis to the main text. See the new section called "Extensions to the Model." Unfortunately, due to space limitations arising from the additional work that was requested and that has now been added to the revised manuscript, we could not move all of the stability analysis, or any of the other analysis, to the main text. We have therefore opted to leave most of the results in the Supplementary Information. If however either the editor or referee prefer a different arrangement (i.e., including extensions to the model in the SI, and some calculations to the main text), we are willing to consider making further changes.

"The the caption of fig.5 the time interval of simulation should be given."

Thanks. Done.

Referee 3

"This paper is based on an interesting idea: what if synchronizing oscillators, as studied in various version of the Kuramoto model, also move in open space such that they tend to synchronize preferentially with neighboring oscillators, and their velocities are also influenced by their degree of synchronization with neighbors? Such a bidirectional coupling between a Kuramoto model and a collective motion model, is not an entirely new idea (see, e.g. models of so-called rippling waves in colonies of myxobacteria, as in Igoshin et al., PNAS 18, 14913 (2001)), but has never been studied before in any detail."

We thank the referee for alerting us to this paper. We have added a paragraph to our Introduction section in which we cite and discuss it. See paragraph beginning "One possible instance..."

"Unfortunately, in my opinion, the authors fail to deliver a nice exploration of this general problem. Their model is rather specific, and rather remote from both any real situation (I doubt the magnetic particles of Snezhko and Aranson have much to do with the current paper). The model is also far from standard models of collective motion (no self-propulsion, no alignment, no noise)"

We take the referee's point that the extreme simplicity of the model casts some doubt on its generality. To remedy this, we have perturbed our model in various ways to see if the phenomena are indeed as robust and generic as we had hoped.

Firstly, we have explored different choices for the interaction functions $I_{att}(x)$, $I_{rep}(x)$... etc of our model and found that all the states of our system persist. We detail this in the "Genericity" section in the Supplementary Information, where we compute the order parameters S, γ and show they have the same qualitative behavior as in the original model, indicating the same states are realized.

Secondly, as suggested by the referee, we have added the features assumed in 'standard models of collective motion'. Specifically, we added the following features, whose analysis we have included in a new "Extensions to the Model" section in the main text, and in various places in the Supplementary Information.

- (i) **Noise.** We explore the addition of white noise on the phase dynamics, and on the spatial dynamics. We also explore anisotropic spatial noise, and the effects of disordered natural frequencies. In each case, we show which states of our system remain, and which do not, and show plots of the order parameters. Generally, the effects of noise is a 'blurring' of the states. ("Noise and disordered frequencies" subsection)
- (ii) **Self-propulsion and alignment.** We explore swarmalators which self-propel and have an orientation β , allowing them to align. We do this by adding a Vicsek-like dynamics for β .
- (iii) As we discuss, these effects constitute a substantial modification: there are now four state variables and many parameters. A comprehensive analysis is therefore beyond the scope of the present paper. So we confine ourselves to checking if the states of our system are robust to inclusion of simple alignment dynamics. We find they are. (Self-propulsion and alignment subsection)

Having done these robustness checks, and verified that the bulk of our states survive, we believe our results have the potential to be realized in nature or technology. Further evidence for this claim can be seen by the similarity between the states of our system and those found in real-world systems, such as vortices of sperm (a description of which we have added in the Results, subsection 4, "Active phase wave").

"I failed to see whether, on the synchronization part, the oscillators have distributed individual frequencies or not (I concluded that, most likely, the oscillators are identical, at odds with the Kuramoto model, but I am not sure!)."

We thank the referee for alerting us to this oversight. In the original paper, we did indeed consider only identical oscillators. We have included a sentence to make this clear. See paragraph starting "For simplicity, we choose power laws" on p.2. However in the revised manuscript, we also consider distributed natural frequencies, as described in the "Extensions to the Model" section.

"We are served a rather elementary numerical study presenting a catalog of 'phases' that are trivial enough to be amenable to a near-exact solution. The coupling functions are near global-coupling (especially for the attractive interaction), which probably explains the relative simplicity of the 'phases' observed."

Throughout the project, we shared the referee's worry that the global coupling might be too special and could give non-generic results (and we especially agree that the choice of the unit vector attraction kernel should be treated with caution). To explore the issue, we changed the forms of the interaction functions as stated in the aforementioned "Genericity" section. In particular, we changed the unit vector attraction kernel to an exponential with a controllable length scale ν , and plotted the order parameters S, γ for different values of ν . We found the results are insensitive to both the specific functional forms and also their length scales.

To illustrate this point, we have included a plot of the order parameters in the "Genericity" section of the Supplementary Information; see Figure S11 (we would have preferred to include it in the main text, but are already at the limit for the number of Figures).

"On a more specific side, no attempt is made to study/evaluate finite-size effects, the nature of the transition between the 'phases', the robustness against noise, so that, in the end, the results remain rather limited in scope."

1. Finite size effects: We have included a section on finite-size effects in the Supplementary Information in the 'Stability of static async state' section. The section in the Supplementary Information on 'Noisy async' state also contains a finite-size study of this state.
2. Robustness to noise: As mentioned above, we have studied the effects of noise in the new "Extensions to the model" section.

REVIEWERS' COMMENTS:

Reviewer #1 (Remarks to the Author):

An extensive revision of the manuscript has been made so that it follows referee reports including mine. Because of this, I would like to recommend that the authors' paper should be published. However, there are some misspellings and a grammatical error (the third line of Fig. S17's caption). These errors should be corrected by the editorial staff, I hope.

Reviewer #2 (Remarks to the Author):

In the revised version, the authors have substantially extended and expanded their work, addressing adequately the comments of my first referee report.

The new section "Extensions to the model" which provides an extension of the model to 3D, in response to my first report, and robustness of the patterns under the influence of noise and disordered natural frequencies, is a valuable and interesting complement to the previous results in 2D.

The authors have also added an extended discussion of connections to previous studies, and have included new results in the supplemental material, which are helpful for the reader and make the manuscript more understandable for a broad audience. Especially, the new videos nicely demonstrate the swarmalator patterns and their dynamics.

In the present form the manuscript can be accepted for publication.